# Effect of Ghrelin on the Cardiovascular System

**DOI:** 10.3390/biology11081190

**Published:** 2022-08-08

**Authors:** Hiroshi Hosoda

**Affiliations:** 1Division of Hypertension and Nephrology, National Cerebral and Cardiovascular Center Hospital, 6-1 Kishibe-Shimmachi, Suita 564-8565, Osaka, Japan; 2Department of Molecular Pathophysiology, Shinshu University School of Medicine, 3-1-1 Asahi, Matsumoto 390-8621, Nagano, Japan

**Keywords:** ghrelin, des-acyl ghrelin, GHS-R, growth hormone, cardiovascular system, sympathetic activity

## Abstract

**Simple Summary:**

Ghrelin is an octanoylated peptide that was initially isolated from rat and human stomachs in the process of searching for an endogenous ligand to the orphan growth hormone secretagogue receptor (GHS-R), a G-protein-coupled receptor. Exogenous or endogenous ghrelin secreted from the stomach binds to GHS-R on gastric vagal nerve terminals, and the signals are transmitted to the central nervous system via the vagal afferent nerve to facilitate growth hormone (GH) secretion, feeding, sympathetic inhibition, parasympathetic activation, and anabolic effects. Ghrelin also binds directly to the pituitary GHS-R and stimulates GH secretion. Ghrelin has beneficial effects on the cardiovascular system, including cardioprotective effects such as anti-heart failure, anti-arrhythmic, and anti-inflammatory actions, and it enhances vascular activity via GHS-R-dependent stimulation of GH/IGF-1 (insulin-like growth factor-1) and modulation of the autonomic nervous system. The anti-heart failure effects of ghrelin could be useful as a new therapeutic strategy for chronic heart failure.

**Abstract:**

Ghrelin, an n-octanoyl-modified 28-amino-acid-peptide, was first discovered in the human and rat stomach as an endogenous ligand for the growth hormone secretagogue receptor (GHS-R). Ghrelin-GHS-R1a signaling regulates feeding behavior and energy balance, promotes vascular activity and angiogenesis, improves arrhythmia and heart failure, and also protects against cardiovascular disease by suppressing cardiac remodeling after myocardial infarction. Ghrelin’s cardiovascular protective effects are mediated by the suppression of sympathetic activity; activation of parasympathetic activity; alleviation of vascular endothelial dysfunction; and regulation of inflammation, apoptosis, and autophagy. The physiological functions of ghrelin should be clarified to determine its pharmacological potential as a cardiovascular medication.

## 1. Introduction

Ghrelin is a physiologically active peptide that was first discovered in human and rat gastric tissues by the author of this article and colleagues in 1999 as an endogenous ligand specific for the growth hormone secretagogue receptor (GHS-R), which had previously been known as an orphan G-protein-coupled receptor [1]. In addition to causing growth hormone (GH) secretion, ghrelin acts on the cardiovascular system and plays roles in promoting feeding, increasing body weight, regulating energy balance, and stimulating gastric acid secretion and gastrointestinal peristalsis (Figure 1). However, the molecular mechanisms of ghrelin’s action on the cardiovascular system and its specific application to treatment have not yet been fully established. This article outlines the currently known cardiovascular effect of ghrelin.

## 2. Characteristics of Ghrelin

Human ghrelin is composed of 28-amino acid residues in which n-octanoic acid (C8:0), a medium-chain fatty acid, is esterified to the side chain of the third serine residue (Figure 2) [1]. In humans, multiple ghrelin-derived molecules with different numbers of fatty acids are generated by post-translational processing [2]. This fatty acid modification is critical for the GHS-R-mediated physiological effects of ghrelin, and is catalyzed by ghrelin o-acyltransferase (GOAT) [3]. Ghrelin is found in a wide range of living organisms, including lower animals such as cartilaginous fish, and in many species, it is produced mainly in the stomach [4]. Gastric ghrelin-producing cells are closed-type cells that do not reach the lumen; they were originally referred to as endocrine A-like cells in the gastric corpus mucosa layer and had unknown functions [5,6]. In the rat stomach, ghrelin gene expression is very low at birth, starts to increase at around 1 week of age, and reaches almost adult levels at 5 weeks of age, prior to puberty [7]. Low levels of ghrelin gene expression are also observed in the small intestine, pancreas, hypothalamus, and other organs. In the gastric tissues and blood, the level of des-acyl ghrelin, the molecular form without fatty acid modification, is higher than that of acyl ghrelin [8]. It is presumed that des-acyl ghrelin is derived from unmodified fatty acids in the stomach tissues or is formed when fatty acids are cleaved from acyl-ghrelin in the blood [9].

In plasma, fatty acid-modified ghrelin occurs at picomolar concentrations, and it is therefore unlikely that it directly acts on peripheral tissues. Plasma ghrelin levels are higher in gastric veins than in peripheral veins [10]. Therefore, ghrelin secreted from gastric ghrelin-producing cells is thought to act directly on ghrelin receptors on vagal nerves distributed throughout the stomach [6].

## 3. Ghrelin Receptor

The ghrelin receptor (GHS-R) was discovered in 1996 by expression cloning [11]. The human GHS-R gene is sited on chromosome 3q26.2, is still conserved across species, and encodes two splicing variants, GHS-R1a and GHS-R1b [12,13]. GHS-R1a cDNA encodes the primary ghrelin receptor, which is a G-protein-coupled receptor containing 366 amino acid residues and 7 transmembrane domains. GHS-R1b cDNA encodes a receptor that is considered functionally inactive and that contains 289 amino acid residues with 5 transmembrane domains; these exhibit deletion of the C-terminal domains, which differentiates them from GHS-R1a. In humans, high GHS-R mRNA expression is observed in the pituitary gland, hypothalamus, and hippocampus [14]. One study reported ghrelin receptor expression in the heart and vascular system [15]. A second study using a [^125^I-His9]-ghrelin-binding assay showed a wide distribution of GHS-R in the human saphenous vein, coronary arteries, left ventricle, and right atrium [16]. By contrast, a study employing a reverse transcription polymerase chain reaction assay reported that GHS-R1a mRNA was not detected in the rat vascular system, and another study using reporter mice detected no GHS-R expression in the aorta, mesenteric arteries, cerebral arteries, coronary arteries, or myocardial tissues [17]. It has been shown that a ghrelin receptor subtype that is different from GHS-R1a and GHS-R1b exists in H9C2 myocardial and endothelial cells [18]. In addition, it is known that the class B scavenger receptor CD36 (84-kD glycoprotein) exists in the circulatory tissues of rats and humans and binds to the ghrelin agonist hexarelin [15]. However, the relationship between the CD36 receptor and ghrelin remains unknown. Further research is needed to determine which receptors for ghrelin and its analogs are active in the cardiovascular system.

Des-acyl ghrelin has no GHS-R binding affinity and exhibits a lack of GH secretion. However, des-acyl ghrelin-specific effects have been demonstrated [19]. Ghrelin and des-acyl ghrelin were shown to protect against skeletal muscle atrophy in a GHS-R1a-independent manner [20]. Further effects will presumably discover des-acyl ghrelin-specific receptors.

## 4. Mechanism of Ghrelin on GH Secretion and Feeding

Peripherally or centrally administered ghrelin activates GH secretion, strongly stimulating feeding behavior in rats and mice [21,22]. This stimulatory effect is also observed with the peripheral administration of ghrelin in humans. Ghrelin is currently the only known peptide that promotes appetite and feeding behavior when administered peripherally. Ghrelin cells have also been observed in the hypothalamic arcuate nucleus, and intracerebroventricular injection of ghrelin antibodies inhibits feeding. Therefore, centrally acting ghrelin also controls feeding.

Peripheral ghrelin signals that stimulate GH secretion and feeding are transmitted to the central nervous system via vagal afferent pathways [6,23]. Ghrelin receptors are synthesized in the cervical ganglia of the afferent vagal nerve and are axonally transported to the vagal nerve endings in the stomach. Ghrelin secreted from gastric endocrine cells into the blood binds to adjacent ghrelin receptors and transmits information to the medulla oblongata by inhibiting the electrical activity of the afferent vagal nerve. This information is conveyed to GH-releasing neurons and neuropeptide Y neurons via the noradrenaline neuron system in the solitary nucleus in the medulla oblongata. Signals are also conveyed to peripheral tissues through the efferent autonomic nervous system, and regulate the following: gastric acid production and gastrointestinal motility via the efferent vagal nerve; blood glucose and lipid metabolism in the liver; and circulatory and brown adipose tissue activities via the sympathetic nervous system [24,25,26,27,28].

## 5. Effects of Ghrelin on the Cardiovascular System

### 5.1. Acute Effects of Ghrelin Administration

#### 5.1.1. Effects of Ghrelin through the Sympathetic Nervous System

In humans and rats, peripherally administered ghrelin significantly reduced mean arterial pressure without heart rate changes [29,30]. Intracerebroventricular administration of ghrelin lowered blood pressure, mildly reduced heart rate, and inhibited renal sympathetic activity in rabbits [31,32]. Continuous intracerebroventricular administration of a non-hypotensive dose of ghrelin increased arterial baroreceptor sensitivity [33]. Injection of a very small dose of ghrelin into the solitary nucleus in the medulla oblongata of rats also lowered blood pressure, reduced heart rate, and inhibited renal sympathetic activity [34]. These responses were inhibited by pretreatment with a ganglion blocking agent. These findings suggest that ghrelin modulates blood pressure by acting on the sympathetic nervous system.

#### 5.1.2. Effects of Ghrelin via GH/IGF-1

GH and IGF-1 (insulin-like growth factor-1) have been reported to increase myocardial contraction, stimulate myocardial remodeling, and promote nitric oxide-dependent vasodilation [35,36,37]. Ghrelin is expected to act on the cardiovascular system via the ghrelin-GH/IGF-1 pathway because it stimulates GH and IGF-1 secretion [38]. In a rat model of chronic heart failure, ghrelin elevated blood levels of GH and IGF-1 and improved cardiac function [30]. Intravenous ghrelin administration also decreased blood pressure in GH-deficient rats [39]. Therefore, the hypotensive action of ghrelin seems to be exerted via both GH/IGF-1 pathway and the autonomic nervous system.

#### 5.1.3. Direct Cardiovascular Effects of Ghrelin

Since ghrelin receptors are expressed in myocardial and vascular smooth muscle cells, the direct cardiovascular effects of ghrelin have been investigated. An early study indicated that the ghrelin and GHR-R1a genes are expressed in human umbilical vein endothelial cells (HUVECs), and that ghrelin induces the growth of HUVECs by inhibiting fibroblast growth factor-2 [40]. Ghrelin relaxed isolated blood vessels that were pre-constricted with endothelin [41]. In a study to investigate the hypotensive effect of intravenous injection of ghrelin in rats, chronic premedication with a nitric oxide synthesis inhibitor enhanced the hypotensive effect of ghrelin [42]. This is because the acute hypotensive response to ghrelin may be exaggerated by chronic premedication with L-NAME, thereby blunting the nitric oxide system. In contrast, administration of apamin/charybdotoxin, which inhibits potassium channels that open when intracellular calcium ion levels increase, inhibited the hypotensive effect of ghrelin, suggesting that ghrelin exerts a vasodilating effect via calcium-sensitive potassium channels during vascular endothelial dysfunction. In GHS-R1a knockout mice, the activity of adenosine monophosphate-activated protein kinase was notably suppressed in vascular endothelial cells [43].

However, some studies reported that ghrelin does not directly affect the cardiovascular system [17,44,45]. The reasons for these conflicting findings may include the fact that the ester linkage of acyl modification of ghrelin is unstable [9,46]. The activity of des-acyl may also contribute, because ghrelin is promptly deacylated in blood and culture solutions. Future studies using GH secretagogues and des-acyl ghrelin should investigate whether ghrelin’s actions depend on GHS-R.

### 5.2. Effects of Chronic Ghrelin Administration

#### 5.2.1. Effects on Acute Myocardial Infarction

In a rat model of myocardial infarction produced by left coronary ligation, cardiac sympathetic activity started gradually increasing from 30 min to 5 h after ligation [47]. On the other hand, when ghrelin was administered immediately after ligation, this sympathetic activity was almost completely inhibited. Interestingly, ghrelin administration 2 h after ligation inhibited sympathetic activity that had been increasing up to that point. Although approximately 61% of the rats died by 6 h after ligation, mortality was markedly decreased by ghrelin administration in the acute phase of myocardial infarction immediately after ligation. Ghrelin seemingly exerted this effect inhibiting ventricular arrhythmias. In a study in which ghrelin (100 µg/kg, twice daily, for 2 weeks) was subcutaneously administered to myocardial infarction model rats beginning on the day after ligation, the increase in left ventricular end-diastolic diameter and decrease in left ventricular systolic performance were significantly mitigated in the ghrelin-treated group compared with controls, although there was no difference in cardiac weight or infarction size [48]. In addition, ghrelin administration reduced the degree of fibrosis in infarcted areas and associated collagen gene expression, and inhibited the progression of cardiac remodeling. Other studies also reported that ghrelin significantly reduced the incidence of ventricular tachyarrhythmia in myocardial infarction model rats [49,50,51].

#### 5.2.2. Effects on Heart Failure

When ghrelin (100 µg/kg, twice daily, for 3 weeks) was subcutaneously administered after left coronary artery ligation in a rat model of chronic heart failure, the blood levels of GH and IGF-1 increased, as did the maximum rate of pressure rise (max dp/dt), which is an index of cardiac output and myocardial contractility [30]. In addition, peripheral vascular resistance was significantly decreased. Therefore, an increase in the cardiac output induced by ghrelin may be partially associated with reduced cardiac afterload resulting from ghrelin’s vasodilating effect. Furthermore, ghrelin induced compensated cardiac hypertrophy in non-infarcted regions as seen in GH replacement therapy in humans. These model rats also showed significant increases in body weight and skeletal muscle mass.

#### 5.2.3. Effects of Ghrelin on Alleviating Heart Failure in Humans

In patients with chronic heart failure, repeated intravenous administration of ghrelin (2 µg/kg, twice daily, for 3 weeks) increased cardiac and skeletal muscle mass, improved cardiac function and exercise capacity, inhibited sympathetic activity, and increased feeding [52]. In heart failure patients with malnutrition, plasma ghrelin levels increased as a compensatory response. It seems that the contributions of ghrelin to regulating energy metabolism also beneficially affect heart failure.

### 5.3. Role of Endogenous Ghrelin

In ghrelin-knockout (KO) and wild-type (WT) mice, myocardial infarction resulted from ligation of the left anterior descending coronary artery, and the conditions of both mouse types were compared 2 weeks after ligation [53]. Compared with WT mice, KO mice had a significantly higher left ventricular weight to tibial length ratio, blood epinephrine levels, and end-systolic and end-diastolic volumes. Additionally, their mortality rate was significantly higher and their systolic function, as represented by ejection fraction was significantly worse. The low-frequency to high-frequency ratio, which is an index of cardiac sympathetic activity, was significantly higher in KO mice. In addition, ghrelin or a β-blocker was administered to KO mice after myocardial infarction to investigate the mechanism of ghrelin activity. Mice treated with ghrelin showed lower mortality than those treated with a β-blocker, although both groups showed a comparable degree of improvement in chronic-phase cardiac function. In a study in which transverse aortic constriction was induced in KO mice, ghrelin inhibited cardiac hypertrophy through activation of the cholinergic anti-inflammatory pathway [54]. Furthermore, in GHS-R knockout mice, endogenous ghrelin could protect against isoproterenol-induced cardiac fibrosis [55]. These findings suggest that endogenous ghrelin alleviates myocardial impairment by inhibiting sympathetic activity and activating parasympathetic activity.

### 5.4. Effects of Des-Acyl Ghrelin on the Cardiovascular System

Des-acyl ghrelin was reported to suppress the apoptosis of myocardial and vascular endothelial cells. Recently, it was shown that des-acyl ghrelin improved cardiac function in the settings of cardiotoxicity induced by the anticancer drug doxorubicin and myocardial impairment associated with cachexia [56]. It is believed that the receptor mediating des-acyl ghrelin-dependent effects are distinct from GHS-R.

## 6. Role of Ghrelin in Cachexia in Patients with Heart Failure

Cardiac cachexia, characterized by severe weight loss caused by heart failure, disturbs the balance between catabolism and anabolism, induces hormonal changes, stimulates cytokine activation, and decreases the mass of skeletal muscles and bones throughout the body [57,58,59,60]. Ghrelin secretion regulates the compensatory response to the lack of energy associated with cachexia [61,62]. Ghrelin is potently adipogenic and orexigenic and promotes weight gain and fat accumulation by increasing both body fat mass and lean tissue mass [63]. Resistance to the appetite-promoting efficacy of ghrelin in heart failure patients was reported as a possible pathogenic factor underlying cardiac cachexia [64]. Cachexic heart failure patients show significantly higher plasma ghrelin levels than healthy individuals and non-cachexic heart failure patients [52]. These elevated levels seem to be a compensatory response to the imbalance between catabolism and anabolism in cardiac cachexia. Exogenous ghrelin is beneficial for cardiac cachexia patients because it increases cardiac output and decreases systemic vascular resistance [65,66]. In addition, exogenous administration of ghrelin was shown to cause strong and persistent stimulation of feeding behavior that was mediated by neuropeptide Y neurons in the hypothalamic arcuate nucleus [22,67]. The ghrelin analogs BIM-28131 and BIM-28125 increased body weight and appetite by regulating the expressions of MuRF-1, MAFbx, and myostatin in experimental heart failure models [68,69]. Elevated plasma ghrelin levels are likely to play a compensatory physiological role in cachexic heart failure patients when the balance between catabolism and anabolism is disrupted.

## 7. Conclusions

Ghrelin promotes GH and IGF-1 functions, inhibits sympathetic activity, activates parasympathetic activity, and improves cardiac function associated with heart failure by regulating energy metabolism. The finding that ghrelin positively impacts heart failure via a mechanism that differs from that of conventional heart failure drugs may be useful for establishing an alternative treatment approach to chronic heart failure. The pharmacological potential of ghrelin pathway modulation should be investigated in more depth to develop new therapeutic options for cardiovascular diseases.

## Figures and Tables

**Figure 1 biology-11-01190-f001:**
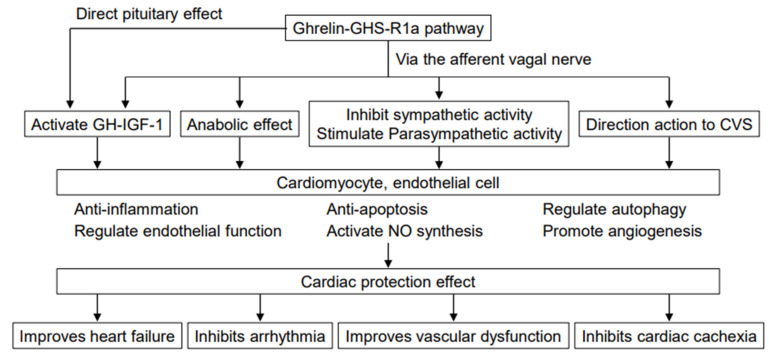
Effect of ghrelin-GHS-R1a pathway on the cardiovascular system. Ghrelin affects the cardiovascular system in many different ways, including inhibiting sympathetic activity, activating parasympathetic activity, improving energy balance, and facilitating mechanisms mediated by GH and IGF-1, thereby improving cardiac function in heart failure. Some of the effects may include direct actions on the cardiovascular system. CVS, cardiovascular system.

**Figure 2 biology-11-01190-f002:**
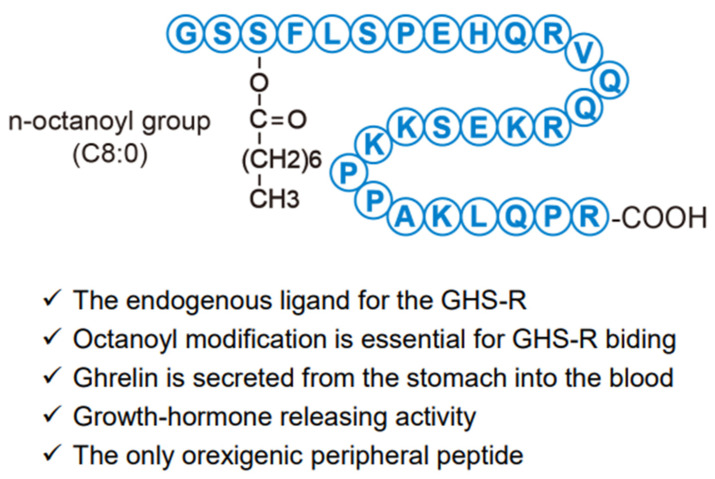
Structure and biological significance of human ghrelin.

## Data Availability

Not applicable.

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
