# Peer review of "Effect of Ghrelin on the Cardiovascular System"

_biology, 2022, doi:10.3390/biology11081190_

Round 1
Reviewer 1 Report
The author of this review is a pioneer in the field of ghrelin physiology. The review considers the central and peripheral effects of ghrelin on the cardiovascular system in a healthy body, as well as in some pathologies (myocardial infarction and heart failure). The topic of the review is new and relevant, it potentially contributes to the creation of new approaches to the treatment of diseases of the cardiovascular system. However, I believe that some changes should be made before the review is accepted for publication.
1) The Figure is not informative enough and not well organized:
- vagal afferents, which, according to the author, are the main target of ghrelin, are not shown;
- "anabolic effect" should be located next to "activate GH-IGF-1";
"activate NO synthesis" should be placed next to "regulate endothelial function";
- the term «enhance vascular activity» is not clear;
- «improve» should be replaced by «improves», «inhibit» should be replaced by «inhibits», etc.
2) line 67: GHR-R1b from GHSR-1a – probably, needs correction
3) lines 74-75: more information on “non-canonical” ghrelin receptor would be useful
4) section 3: the title does not correspond to the content. Correct, please.
5) lines 115-116: These responses were inhibited by premedication with a sympatholytic. In my opinion, sympatholytic drugs have no direct effects on efferent (renal) sympathetic nerve activity.
6) lines 134-135: “premedication with a nitric oxide synthesis inhibitor enhanced the hypotensive effect of ghrelin”. The mechanism is not clear, explain, please.
7) section 4.2.1. should appear after 4.2.2. (before 4.2.3.)
8) lines 189-190: The ratio of low frequency to high frequency, which is an index of sympathetic activity. Probably, you describe the effects of ghrelin on low/high frequency oscillations of heart rate - ?.
9) lines 195-196: ghrelin inhibited 195 the chronic inflammatory response of the heart mediated by the parasympathetic nervous 196 system. The sentence should be corrected.

Reviewer 2 Report
Hosoda reviewed recent progresses in the molecular mechanisms of ghrelin on the cardiovascular system. The manuscript focuses on functions of ghrelin, emphasizes the pharmacological potential of ghrelin in the treatment of cardiovascular system diseases. This manuscript provides a different perspective than a similar review article written by Takeshi Tokudome in 2019. The review is well-written and I only have minor comments listed below:
1. In the section of 4.1.3., please include discussion of ghrelin’s role in activation of myocardial GHSR and the downstream signaling pathways that lead to cardioprotective effects. Recently, Sullivan et al (2021) reported a positive correlation between GHSR and ghrelin only in diseased state.
2. In the section of 4.3, when discuss the chronic inflammatory response of the heart, it is worthwhile to point out that ghrelin could inhibit inflammatory response and apoptosis of myocardial injury in septic rats through JAK/STAT signaling pathway (Jin et al. 2020), alter arterial stiffness (Ali et al. 2017), suppress the AMI-induced inflammatory response, and inhibits the activation of autophagy (Yuan et al. 2020). On the other hand, GHSR deficiency exacerbates cardiac fibrosis (Gupta et al. 2020).
3. In the section of 4.3., it is worth mentioning that LEAP2 has been reported as an endogenous antagonist of the ghrelin receptor (Ge et al. 2018), a possible regulatory mechanism to fine-tuning sympathetic activity and parasympathetic activity.
4. In Figure 1, please define CVS as cardiovascular system.
5. It is worthwhile to add a section with discussions about the pharmacological potential of ghrelin in cardioprotection. Beyond the proposal to use ghrelin and ghrelin/total cholesterol ratio as predictors for coronary artery disease (Niknam et al. 2022), several studies have already shed light on the pharmacological potential of ghrelin in cardioprotection. For example, preconditioning of mesenchymal stem cells with ghrelin exerts cardioprotection in aged heart (Sun et al. 2021); pretreatment of ghrelin protects H9c2 cells against hypoxia/reoxygenation-induced cell death via PI3K/AKT and AMPK pathways (Chen et al. 2019); ghrelin therapy improves lung and cardiovascular function in experimental emphysema (Rocha et al. 2017); nanogel-based nasal ghrelin vaccine prevents obesity (Azegami et al. 2017). Also an adeno-associated virus vector with ghrelin has been developed, which preserves cardiac function over time when delivered to the heart (Ruozi et al. 2015).
Reviewer 3 Report
The paper is devoted to an interesting and quite important problem – studying the functions of ghrelin in the cardiovascular system and the pharmacological potential of the substance in the treatment of cardiovascular diseases.
In the paper the effect of a ghrelin, which was identified as an endogenous ligand for the growth hormone secretagogue receptor, on the cardiovascular system is characterised.
The question of the role of ghrelin in the organism has growing interest, as Google scholar search shows around 17 200 results for the last 5 years with key words “Ghrelin and cardiovascular system”.
The structures of the human ghrelin and ghrelin receptor are characterised. Effects of ghrelin on blood pressure through the sympathetic nervous system are described.
The pathways of ghrelin acting on cardiovascular system are proposed. Direct effects of ghrelin on the cardiovascular system in studies on rats and mice as model organisms are shown. Effect of chronic ghrelin administration in a rat model of chronic heart failure is described. The effects of ghrelin during the acute phase of myocardial infarction in a rat model of myocardial infarction are shown. On the basis of the analysis of the data Author made a conclusion that ghrelin might beneficially affect heart failure while regulating energy metabolism of the organism. The roles of endogenous ghrelin during myocardial infarction in ghrelin-knockout and wild-type mice are analysed. Role of ghrelin in cachexia in patients with heart failure is shown.
The following comments do not diminish the value of the article and have to be considered mostly as recommendations:
The text of the paper is around 3000 words, while it is recommended in the instructions for Authors that the main text of review papers should be around 4000 words and should include at least two figures or tables.
It would be interesting, if the paper would be added with the information about ghrelin functions in promoting vascular activity and angiogenesis if possible. If a scheme which would demonstrate the molecular mechanism of ghrelin action on the cardiovascular system with the existing questions indicated would be added, if possible, it would make the material more illustrative.
Lines 4, 5 It would be better to represent the information about the Author and affiliations according the requirements of the Journal.
Figure 1 It would be better to decode an abbreviation CVS.
Line 31 Probably it would be better to mention once the title of the Figure 1?
Line 149, line 178 It would be better to add “BID” abbreviation meaning.
Line 235 It would be better to remove additional number of each reference.
